# Effect of metformin monotherapy on serum lipid profile in statin-naïve individuals with newly diagnosed type 2 diabetes mellitus: a cohort study

Szu Han Lin[1,*], Po Chung Cheng[1,*], Shih Te Tu[1], Shang Ren Hsu[1], Yun Chung Cheng[2] and Yu Hsiu Liu[3]

[1] Division of Endocrinology and Metabolism, Department of Internal Medicine, Changhua Christian Hospital, Changhua City, Taiwan
[2] Department of Radiology, Taichung Veterans General Hospital, Taichung, Taiwan
[3] Department of Accounting and Information Systems, National Taichung University of Science and Technology, Taichung, Taiwan
[*] These authors contributed equally to this work.

## ABSTRACT

**Background**. Cardiovascular disease is a major cause of mortality and morbidity in people with type 2 diabetes mellitus (T2DM). Studies have consistently identified dyslipidemia as an important risk factor for the development of macrovascular disease. The landmark United Kingdom Prospective Diabetes Study has shown that metformin therapy reduces cardiovascular events in overweight people with T2DM. This study investigates the effect of metformin monotherapy on serum lipid profile in statin-naïve individuals with newly diagnosed T2DM, and whether the effect, if any, is dosage-related.
**Methods**. This cohort study enrolled individuals exceeding 20 years of age, with recent onset T2DM, who received at least 12 months of metformin monotherapy and blood tests for serum lipid at 6-month intervals. Exclusion criteria involved people receiving any additional antidiabetic medication or lipid-lowering drug therapy. Lipid-modifying effect of metformin was recorded as levels of serum triglycerides (TG), high density lipoprotein cholesterol (HDL-C), and low density lipoprotein cholesterol (LDL-C) measured at six month intervals.
**Results**. The study enrolled 155 participants with a mean age of 58.6 years and average glycosylated hemoglobin $A_{1c}$ of 8%. After initiating metformin therapy, LDL-C was significantly reduced from 111 mg/dl to 102 mg/dL at 6 months ($P < 0.001$), TG was reduced from 132 mg/dl to 122 mg/dL at 12 months ($P = 0.046$), and HDL-C increased from 45.1 mg/dL to 46.9 mg/dL at 12 months ($P = 0.02$). However, increasing the dosage of metformin yielded no significant effect on its lipid-lowering efficacy.
**Discussion**. Metformin monotherapy appreciably improves dyslipidemia in statin-naive people with T2DM. Its lipid-modifying effect may be attributable to insulin sensitization, reduction of irreversibly glycated LDL-C, and weight loss. In practice, people with dyslipidemia who are ineligible for lipid-lowering agents may benefit from metformin therapy. Moreover, previous studies report a synergistic effect between metformin and statin, which may further reduce cardiovascular events in at-risk individuals. Overall, metformin is a safe and efficacious approach to alleviate dyslipidemia in people with newly diagnosed T2DM.

Corresponding author
Shang Ren Hsu, 67781@cch.org.tw, wintry_morn@msn.com

## INTRODUCTION

Cardiovascular disease causes substantial morbidity in people with type 2 diabetes mellitus (T2DM) (*Kannel & McGee, 1979*). Coronary heart disease not only leads to an increased mortality rate (*Martín-Timón et al., 2014*) but also contributes to long-term cardiac dysfunction (*Miki et al., 2013*). Studies have consistently identified dyslipidemia as an important risk factor for the development of macrovascular disease in T2DM (*Schofield et al., 2016*; *Stern, 1995*; *Wu & Parhofer, 2014*). A safe and efficacious intervention for diabetic dyslipidemia is necessary to attenuate cardiovascular disease in at-risk individuals.

Metformin is the first-line treatment for people with newly diagnosed T2DM endorsed by the *American Diabetes Association (2017)*. This medication primarily inhibits hepatic gluconeogenesis, thereby reducing fasting glucose levels (*Song, 2016*). Moreover, metformin is an antioxidant that diminishes cancer risk (*Kasznicki, Sliwinska & Drzewoski, 2014*) and improves insulin sensitivity (*Shaw, 2013*). Importantly, the landmark United Kingdom Prospective Diabetes Study (UKPDS) has shown that metformin therapy reduces cardiovascular events in overweight people with T2DM (*American Diabetes Association, 2002*).

Clinical trials have given a hint to the lipid-modifying effect of metformin. In people with recent onset T2DM, metformin enhances the lipid-lowering efficacy of atorvastatin (*Kashi et al., 2016*). A recent metabolomics study demonstrates that metformin may reduce low density lipoprotein cholesterol (LDL-C) via an AMP-activated protein kinase pathway (*Xu et al., 2015*). However, none of the published studies specifically excludes individuals receiving statin therapy or second-line antidiabetic medications in addition to metformin. Considering the predominant lipid-lowering effect of statin in people with T2DM (*Colhoun et al., 2004*), potential lipid-modifying capacity of metformin is difficult to isolate from that of concomitant statin therapy.

This study investigates the effect of metformin monotherapy on serum lipid profile in statin-naïve individuals with newly diagnosed T2DM. In addition, the study determines whether metformin's lipid-modifying effect, if any, is dosage-related in these participants.

## MATERIALS AND METHODS

This cohort study was conducted at Changhua Christian Hospital in central Taiwan. Participants who visited the Endocrinology clinic between December 2013 and November 2015 were screened for eligibility. Inclusion criteria were individuals exceeding 20 years of age, with recent onset T2DM, who received at least 12 months of metformin monotherapy and blood tests for serum lipid at 6-month intervals. Exclusion criteria involved people receiving second-line antidiabetic medications or lipid-lowering drugs such as statin, fibrate, bile acid sequestrant, nicotinic acid, or ezetimibe. Individuals with familial hypercholesterolemia, thyroid disorder, renal dysfunction, or alcoholism were also ineligible. The study was approved by the Institutional Review Board of Changhua Christian

**Table 1 Demographic features of participants at diagnosis of type 2 diabetes mellitus.**

| Variables | Study population ($n = 155$) |
|---|---|
| Age (years) | 58.6 ± 13.4 |
| Sex (Female) | 77 (49.7%) |
| $HbA_{1c}$ (%) | 8.0 ± 1.8 |
| Creatinine (mg/dL) | 0.84 ± 0.22 |
| ALT (U/mL) | 38.4 ± 27.0 |
| Systolic blood pressure (mm Hg) | 133 ± 16.3 |
| Metformin dose (mg per day) | 1,487 ± 433.8 |
| Microalbuminuria (mg per day) | 51.1 ± 149 |
| Triglycerides (mg/dL) | 132 ± 71.9 |
| High density lipoprotein cholesterol (mg/dL) | 45.1 ± 12.0 |
| Low density lipoprotein cholesterol (mg/dL) | 111 ± 32.3 |

**Notes.**

Data are expressed as mean with standard deviation for continuous variables and number (%) for categorical variables.
ALT, alanine aminotransferase; $HbA_{1c}$, glycosylated hemoglobin $A_{1c}$.

Hospital (CCH IRB: 180102). All participants provided written consent to take part in the study.

Demographic information including age, sex, and systolic blood pressure were recorded at diagnosis of T2DM. Thereafter participants received blood tests for serum lipid at 6-month intervals. Metformin dosage for each individual was defined as the daily quantity received for the longest duration in the first year of treatment. Lipid-modifying effect of metformin was recorded as levels of serum triglycerides (TG), high density lipoprotein cholesterol (HDL-C), and low density lipoprotein cholesterol (LDL-C) measured at six month intervals. The lipid tests in this study were performed using Beckman Coulter UniCel DxC 800 Synchron™ Clinical Systems. The analytical precision was within 1.7 mg/dL for HDL-C, within 3.0 mg/dL for LDL-C, and within 7.5 mg/dL for TG.

Paired $t$-test enabled comparison of participants' serum lipid profile at diagnosis and after receiving metformin therapy. Levels of serum lipid were compared between dosage-based subgroups using one-way analysis of variance. A two-tailed $P$ value of less than 0.05 indicated statistical significance. Statistical analysis was performed using IBM SPSS version 22.0 (IBM SPSS Statistics for Windows. Armonk, NY, USA).

## RESULTS

The study screened 180 individuals for eligibility. Eighteen patients were excluded due to prescription of second-line antidiabetic medications, and seven were ineligible because they received lipid-lowering drugs. The enrollment process is illustrated in Fig. 1.

The study enrolled 155 participants whose demographic features are summarized in Table 1. They had a mean age of 58.6 years and an average glycosylated hemoglobin $A_{1c}$ ($HbA_{1c}$) of 8%. In terms of serum lipid profile, participants harbored a mean LDL-C level of 111 mg/dL, mean HDL-C of 45.1 mg/dL, and mean TG of 132 mg/dL.

After initiating metformin therapy, LDL-C was significantly reduced from 111 mg/dl to 102 mg/dL at 6 months ($P < 0.001$), TG was reduced from 132 mg/dl to 122 mg/dL

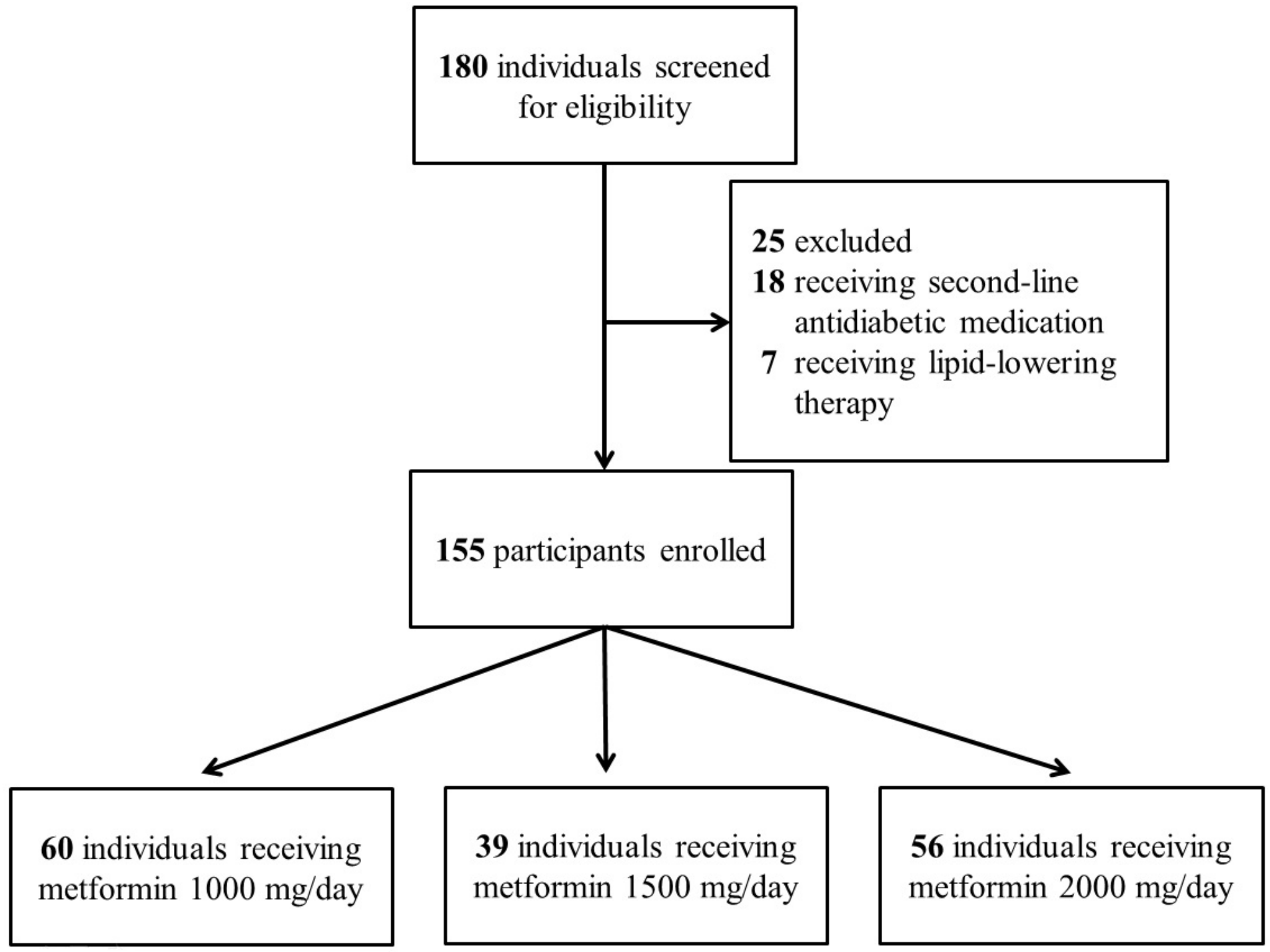

**Figure 1 Enrollment protocol for the study.** Number of participants enrolled in the study and reason for exclusion.

at 12 months ($P = 0.046$), and HDL-C increased from 45.1 mg/dL to 46.9 mg/dL at 12 months ($P = 0.02$). After 12 months of treatment, mean $HbA_{1c}$ was reduced from 8% to 6.47% ($P < 0.001$), and mean body weight changed from 69 kilograms to 68.3 kilograms ($P = 0.01$). These findings are tabulated in Table 2.

As shown in Table 3, increasing the dosage of metformin had no significant effect on its lipid-lowering efficacy. Specifically, improvement in TG, HDL-C, and LDL-C was similar between participants receiving 1,000 mg, 1,500 mg, and 2,000 mg of metformin per day. At the end of this study, all three subgroups had comparable mean $HbA_{1c}$, while those receiving 2,000 mg of metformin per day harbored higher mean body weight.

## DISCUSSION

Metformin may counter the derangements in lipid metabolism in T2DM through several pathways (*Malin et al., 2012*; *Han & Kaufman, 2016*). Through increasing insulin sensitivity, metformin reduces the rate of lipolysis, thereby slowing the conversion of free fatty acids to lipoprotein precursors in the liver (*Melmed et al., 2016*). By reducing plasma glucose levels, metformin lowers the fraction of irreversibly glycated LDL-C, which is removed less efficiently from the body (*Sima et al., 2010*). Metformin also improves dyslipidemia by inducing weight loss in people with impaired glucose metabolism (*Diabetes Prevention Program Research Group, 2012*; *Harder, Dinesen & Astrup, 2004*). Following

**Table 2 Clinical effect of metformin in people with newly diagnosed type 2 diabetes mellitus.**

| Treatment duration (months) | Study population ($n = 155$) | *P* value |
|---|---|---|
| Triglycerides (mg/dL) | | |
| 0 | 132 ± 71.9 | |
| 6 | 123 ± 66.2 | 0.055 |
| 12 | 122 ± 63.6 | 0.046 |
| High density lipoprotein cholesterol (mg/dL) | | |
| 0 | 45.1 ± 12.0 | |
| 6 | 46.3 ± 13.3 | 0.076 |
| 12 | 46.9 ± 13.7 | 0.020 |
| Low density lipoprotein cholesterol (mg/dL) | | |
| 0 | 111 ± 32.3 | |
| 6 | 102 ± 25.4 | 0.001 |
| 12 | 102 ± 25.0 | 0.001 |
| Glycated hemoglobin $A_{1c}$ (%) | | |
| 0 | 8.00 ± 1.83 | |
| 6 | 6.62 ± 0.88 | 0.001 |
| 12 | 6.47 ± 0.68 | 0.001 |
| Body weight (kilograms) | | |
| 0 | 69.0 ± 14.5 | |
| 12 | 68.3 ± 14.4 | 0.010 |

**Notes.**
Data are expressed as mean with standard deviation for continuous variables. Variables are compared to baseline levels using paired *t*-test.

metformin treatment, weight loss is in general modest and attributable to fat loss rather than to energy expenditure (*Yanovski et al., 2011*).

As observed in the current study, metformin monotherapy lowered LDL-C level after only 6 months of therapy, whereas serum TG and HDL-C did not significantly improve until after 12 months. This lipid-modifying effect is concordant with existing evidence that hypertriglyceridemia and diminished HDL-C require a longer therapeutic duration to counteract than lowering LDL-C (*Orchard et al., 2005*). Moreover, metformin's effect on serum lipid appeared to be dosage-independent in this study, suggesting this medication may control dyslipidemia through indirect pathways.

Clinical implications arise from the observation that metformin monotherapy improves dyslipidemia in diabetic patients. Metformin not only controls hyperglycemia but also reduces cardiovascular risk, as shown previously in the UKPDS (*American Diabetes Association, 2002*). Considering the cost and potential side effects of lipid-lowering medications such as statins (*Raymond et al., 2014*), people with dyslipidemia who are ineligible for lipid-lowering therapy may nonetheless benefit from metformin treatment. Previous investigators have also observed a synergistic effect between metformin and statin (*Kashi et al., 2016*), which can further reduce cardiovascular events in at-risk individuals.

**Table 3 Comparison of clinical effect of metformin between dose-based subgroups.**

| Treatment duration (months) | Metformin 1,000 mg/day ($n=60$) | Metformin 1,500 mg/day ($n=39$) | Metformin 2,000 mg/day ($n=56$) | *P* value |
|---|---|---|---|---|
| Triglycerides (mg/dL) | | | | |
| 0 | 131 ± 60.3 | 138 ± 86 | 130 ± 71.9 | 0.854 |
| 6 | 121 ± 71.6 | 128 ± 62.8 | 121 ± 63.5 | 0.841 |
| 12 | 118 ± 58.7 | 129 ± 68.7 | 120 ± 65.7 | 0.717 |
| High density lipoprotein cholesterol (mg/dL) | | | | |
| 0 | 44.9 ± 13.3 | 46.3 ± 10.6 | 44.5 ± 11.5 | 0.748 |
| 6 | 47.0 ± 15.0 | 47.5 ± 13.9 | 44.6 ± 10.9 | 0.510 |
| 12 | 47.0 ± 14.1 | 47.5 ± 15.5 | 46.3 ± 11.8 | 0.910 |
| Low density lipoprotein cholesterol (mg/dL) | | | | |
| 0 | 112 ± 39.2 | 107 ± 28.4 | 111 ± 26.4 | 0.725 |
| 6 | 101 ± 24.8 | 101 ± 27.5 | 105 ± 24.7 | 0.691 |
| 12 | 102 ± 26.8 | 100 ± 21.3 | 103 ± 25.6 | 0.859 |
| Glycated hemoglobin $A_{1c}$ (%) | | | | |
| 0 | 7.76 ± 1.81 | 7.90 ± 1.31 | 8.30 ± 2.13 | 0.276 |
| 6 | 6.78 ± 0.93 | 6.49 ± 0.79 | 6.53 ± 0.89 | 0.180 |
| 12 | 6.55 ± 0.69 | 6.42 ± 0.67 | 6.42 ± 0.67 | 0.527 |
| Body weight (kilograms) | | | | |
| 0 | 64.8 ± 13.2 | 67.4 ± 12.8 | 74.8 ± 15.4 | 0.001 |
| 12 | 64.2 ± 13.9 | 66.4 ± 12.1 | 74.0 ± 15.0 | 0.001 |

**Notes.**
Data are expressed as mean with standard deviation for continuous variables. Variables are compared between dose-based subgroups using analysis of variance.

As demonstrated in this study, metformin modifies serum lipid in a dosage-independent manner. Considering that the gastrointestinal side effect of metformin occurs more frequently at higher dosage (*McCreight, Bailey & Pearson, 2016*), people who experience such side effect may benefit from a lower dosage of metformin, which nonetheless provides lipid-lowering effect. Moreover, in people with adequate glycemic control using second-line antidiabetic medications, adding metformin at low dose may further reduce their cardiovascular risk through the beneficial effect on lipid profile.

This study differs from previous investigations of metformin therapy that did not specifically exclude recipients of lipid-lowering medications, which obviously constitute a confounder on serum lipid profile. The current study reduces potential confounders by excluding recipients of either lipid-lowering drugs or second-line antidiabetic agents. Moreover, the participants have received follow up at the same medical center throughout the study, which circumvents variations in laboratory test results due to different analytic methods.

Several limitations arise from the study design. Therapeutic lifestyle change is an essential component of dyslipidemia treatment (*Mannu et al., 2013*). Since all participants received

self-management education by diabetes educators, a beneficial effect of metformin on serum lipid is difficult to distinguish from that of lifestyle modification. Indeed, the lack of a control group receiving only lifestyle intervention constitutes an important limitation in this study. However, considering that early treatment to glycemic target substantially improves clinical outcome (*American Diabetes Association, 2017*), establishing a control group without any antidiabetic medication may be inappropriate at diagnosis of T2DM.

Another limitation is that serum lipid levels may change as a consequence of anti-hyperglycemic drugs (*Buse et al., 2004*), exercise (*Balducci et al., 2009*), and diet (*Greco et al., 2014*). However, the influence of exercise and diet on serum lipid levels was not addressed by the study. Finally, the mechanism behind metformin's lipid-modifying effect remains elusive, and further studies are required to explore the associated metabolic pathways.

## CONCLUSIONS

In people with newly diagnosed T2DM, metformin therapy significantly reduced both serum LDL-C and TG, as well as raised HDL-C, without concomitant lipid-lowering medications. Moreover, the lipid-modifying effect of metformin appeared to be dosage-independent. Overall, metformin is a safe and efficacious approach to alleviate dyslipidemia in people with T2DM.

### Funding

The authors received no funding for this work.

### Competing Interests

The authors declare there are no competing interests.

### Author Contributions

- Szu Han Lin and Shang Ren Hsu conceived and designed the experiments, performed the experiments, authored or reviewed drafts of the paper, approved the final draft.
- Po Chung Cheng conceived and designed the experiments, performed the experiments, contributed reagents/materials/analysis tools, authored or reviewed drafts of the paper, approved the final draft.
- Shih Te Tu conceived and designed the experiments, contributed reagents/materials/-analysis tools, authored or reviewed drafts of the paper, approved the final draft.
- Yun Chung Cheng analyzed the data, prepared figures and/or tables, authored or reviewed drafts of the paper, approved the final draft.
- Yu Hsiu Liu analyzed the data, prepared figures and/or tables, authored or reviewed drafts of the paper, approved the final draft, as qualified statistician.

### Human Ethics

The following information was supplied relating to ethical approvals (i.e., approving body and any reference numbers):

The study was approved by the Institutional Review Board of Changhua Christian Hospital (CCH IRB: 180102).

## Data Availability

The dataset is available as a Supplemental File.

## Supplemental Information

Supplemental information for this article can be found online at http://dx.doi.org/10.7717/peerj.4578#supplemental-information.

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
