# Peer review of "Effect of metformin monotherapy on serum lipid profile in statin-naïve individuals with newly diagnosed type 2 diabetes mellitus: a cohort study"

_PeerJ, doi:10.7717/peerj.4578_

## Round 0.1 · original submission · Major Revisions

· Academic Editor

Major Revisions

Authors should try to add information about clinical and biochemical features at the end of the study, as pointed out by Reviewer 1, and at least discuss the lack of a control group treated with lifestyle changes only.

Please, improve references. For example, line 81, I would suggest to add: Kannel WB, McGee DL. Diabetes and cardiovascular disease. JAMA (1979) 241:2035–8.

Line 84, please add: Stern MP. Diabetes and cardiovascular disease. The “common soil” hypothesis. Diabetes (1995) 44:369–74.
Wu L, Parhofer KG. Diabetic dyslipidemia. Metabolism (2014) 63:1469–79.

Line 129: please add the lipid tests employed in this study and their analytical performance (precision).

Line 164: following metformin treatment, weight loss is in general modest and it is mostly due to fat loss rather than to reduced energy expenditure. Please, add this information in the text.

Reviewer 1 ·

Basic reporting

No comment

Experimental design

see below

Validity of the findings

See below

Comments for the author

In this paper, the authors demonstrated that metformin monotherapy improves serum lipid profile in statin-naïve individuals with newly diagnosed T2DM.
I have a major comment that needs to be addressed:

Major concern
As underlined by authors, the strong limitation of the present study is the lack of a control group treated with lifestyle changes only. In fact, not only lipid profile changes as a consequence of anti-hyperglycemic drugs (Buse JB, et al. Diabetes Obes Metab 2004), but also the beneficial effect of exercise (Balducci S, et al. Diabetes Metab Res Rev. 2009 Sep;25 Suppl 1:S29-33. doi: 10.1002/dmrr.985) and diet (Greco M, et al. Mediators Inflamm. 2014;2014:750860. doi: 10.1155/2014/750860) on lipid profile is well known. These observations should be added in the discussion (line 204). Also, any information about clinical and biochemical features at the end of the study is lacking. The authors must indicate at least body weight and HbA1c in the three treated cohorts at the end of the study and discuss this point.

Minor concerns
Bibliography should be revised. Authors could have provided something more specific than the whole text by Melmed (line 256), while references at lines 234 and 260 could be substituted, or accompanied by more prominent studies.

Grammar and typo errors deserve further re-check. For example, in line 160, “improved” should be “improve”.

·

Basic reporting

EXCEPT FOR SCATTERED STYLISTIC EDIFICATION _FOR ABIDING BY PEERJ GUIDELINES_ THE STUDY SIGNIFICANCE AND IMPACT OF CLINICAL RELEVANCE OF METFORMIN MONOTHERAPY IN DRUG NAIVE DM2 TO DYSLIPIDEMIA IS NOVEL AND UNPRECEDENTED!

Experimental design

SCIENTIFICALLY SOUND AND COMPREHENSIVE

Validity of the findings

FINDINGS ARE VALIDATING THE HYPOLIPIDEMIC EFFICACY OF METFORMIN MONOTHERAPY ON A TITRATION SCALE_THIS VASTLY REFLECTS ON ITS RE-PURPOSED PHARMACOLOGY AS A HYPOLIPIDEMIC PHARMACOTHERAPEUTIC AGENT

Comments for the author

ACCEPT AFTER MINOR REVISIONS

---

## Round 0.2 · accepted · Accept

· Academic Editor

Accept

The authors have satisfactorily addressed all the issues raised by the Reviewers and the Editor.

Reviewer 1 ·

Basic reporting

OK

Experimental design

OK

Validity of the findings

OK

Comments for the author

Accordingly with my requests, the paper has been improved and it is now susceptible for publication.